# Evaluation of the Cardiometabolic Disorders after Spinal Cord Injury in Mice

**DOI:** 10.3390/biology11040495

**Published:** 2022-03-24

**Authors:** Adel B. Ghnenis, Calvin Jones, Arthur Sefiani, Ashley J. Douthitt, Andrea J. Reyna, Joseph M. Rutkowski, Cédric G. Geoffroy

**Affiliations:** 1Department of Neuroscience and Experimental Therapeutics, College of Medicine, Texas A&M Health Science Center, Bryan, TX 77807, USA; abghneni@med.umich.edu (A.B.G.); bonesjones64@tamu.edu (C.J.); sefiani@tamu.edu (A.S.); ashleydouthitt@tamu.edu (A.J.D.); 2Department of Medical Physiology, College of Medicine, Texas A&M Health Science Center, Bryan, TX 77807, USA; areyna94@tamu.edu (A.J.R.); rutkowski@tamu.edu (J.M.R.)

**Keywords:** spinal cord injury severity, cardiometabolic disease, liver and cardiac dysfunctions, fibrosis

## Abstract

**Simple Summary:**

The present study demonstrates severity-dependent effects of a thoracic T8 injury on cardiometabolic dysfunctions in adult mice. While these chronic cardiometabolic issues can be multifactorial, the data indicate that systemic inflammatory response is likely to be involved. These findings are supportive of the role of systemic inflammation following SCI being critical in identifying therapeutic targets and predicting long-term outcomes.

**Abstract:**

Changes in cardiometabolic functions contribute to increased morbidity and mortality after chronic spinal cord injury. Despite many advancements in discovering SCI-induced pathologies, the cardiometabolic risks and divergences in severity-related responses have yet to be elucidated. Here, we examined the effects of SCI severity on functional recovery and cardiometabolic functions following moderate (50 kdyn) and severe (75 kdyn) contusions in the thoracic-8 (T8) vertebrae in mice using imaging, morphometric, and molecular analyses. Both severities reduced hindlimbs motor functions, body weight (g), and total body fat (%) at all-time points up to 20 weeks post-injury (PI), while only severe SCI reduced the total body lean (%). Severe SCI increased liver echogenicity starting from 12 weeks PI, with an increase in liver fibrosis in both moderate and severe SCI. Severe SCI mice showed a significant reduction in left ventricular internal diameters and LV volume at 20 weeks PI, associated with increased LV ejection fraction as well as cardiac fibrosis. These cardiometabolic dysfunctions were accompanied by changes in the inflammation profile, varying with the severity of the injury, but not in the lipid profile nor cardiac or hepatic tyrosine hydroxylase innervation changes, suggesting that systemic inflammation may be involved in these SCI-induced health complications.

## 1. Introduction

It is estimated that there are 245,000 to 353,000 persons suffering from SCI in the United States, with approximately 17,500 new cases each year [1]. These injured individuals are known to face lifelong locomotor disabilities, resulting from the injury to the central nervous system leading to neural cells loss, axon degeneration, and other cellular events at the injury site [2,3]. What is less known is that people living with SCI experience numerous detrimental medical complications that have great influence on their quality of life, including diabetes and cardiovascular disease [4,5]. Individuals with chronic SCI are prone to cardiometabolic disease [6,7,8,9]. In the general population, risk factors, such as metabolic syndrome (MS), obesity, glucose intolerance, hypertension, high blood triglycerides levels, and decreased high-density lipoprotein (HDL) [10,11], increase the prevalence of cardiovascular disease (CVD) [12,13]. In the general population, more than a third of US individuals suffer from MS [14,15]. MS symptoms are more prevalent in SCI patients than in the general population. In a recent study, it was estimated that more than 50% of SCI patients have symptoms of MS [16]. Increased lipid profile, an important biomarker for MS, induces insulin resistance, systemic inflammation, and oxidative stress [17]. The high prevalence of MS and decreased physical activity affect the balance between energy intake and energy expenditure and predispose these individuals to MS. In this regard, individuals suffering from SCI have reduced physical activity, systemic inflammation, and changes in the distribution pattern of adipose tissue in the body, which can all participate in the development of MS in these population [18,19].

SCI not only effects motor and sensory function within the central nervous system, but also disrupts the peripheral neural circuitry and signals to vital organs in the body, resulting in severe long-term complications outside of the central nervous system [4]. A large body of evidence indicates that SCI has major effects on body composition and metabolism [20,21,22], which can lead to a variety of risk factors including obesity [7,23,24], lower limbs skeletal muscle atrophy [25,26], decreased daily energy expenditure [13], changes in glucose-insulin homeostasis [21,22,27], and cardiovascular disease [28]. Assessment of body composition following SCI is critical to predict the development of cardiometabolic diseases, but is not commonly accessible in clinical evaluations [29,30]. Data from both humans and experimental animals indicate that in the early stages of SCI, body weight is decreased due to the reduction in lean body mass and fat depots [31,32]. These changes depend on the injury severity, level of injury, and duration of the injury [22,33].

The liver is a key organ regulating many metabolic processes in the body, and plays a vital role in body energy metabolism [34]. Under normal physiological conditions, digested food components such as glucose, amino acids, and fatty acids are transported to the liver. The liver metabolizes glucose into pyruvate for ATP production and produces the substrates required to synthesize fatty acids through lipogenesis. These fatty acids can be stored as lipid droplets and membrane structures in the hepatocytes or secreted into blood circulation. During starvation, liver gluconeogenesis induces glucose production and promotes lipolysis. These metabolic pathways are highly regulated through neuronal (the sympathetic and parasympathetic system) and hormonal (insulin and glucagon) pathways [35]. Therefore, liver dysfunction may lead to many complications including type 2 diabetes, insulin resistance, and nonalcoholic fatty liver diseases (NAFLD) [36]. Although SCI increases the prevalence of MS, only a few human studies have examined liver function following SCI. Using ultrasound imaging, Sipski et al. reported that approximately 80% of chronic SCI patients exhibited liver abnormality [37]. A recent study by Rankin et al. used magnetic resonance imaging (MRI) to demonstrate increased liver adiposity during chronic SCI which impacts the metabolic profile, highlighting the critical need to measure liver adiposity following SCI [38]. In a rat SCI model, Sauerbeck et al. reported hepatic changes including increased lipid infiltration and inflammation in the liver as early as 3 weeks PI [39]. The systemic inflammation and increased inflammatory cytokines in the liver following SCI may significantly induce liver dysfunction [40,41]. Understanding the progression of metabolic diseases could help identify points of intervention to increase the life expectancy and quality of life of individuals with SCI.

CVD is the leading cause of mortality among SCI individuals [42]. Depending on the severity and injury level, SCI may disrupt the innervation to the heart, causing cardiovascular autonomic dysfunction, leading to blood pressure and heart rate dysregulation [43,44,45,46]. In humans, patients with cervical and high-thoracic level injuries are more likely to develop CVD [47], probably due to the change in sympathetic nervous activity. For this reason, most of the data from experimental animals are focused on the SCI above the sixth thoracic vertebra (T6) [48]. This includes T5 complete transection in rats [49,50] and T3 moderate or severe contusion in rats [51,52] which can lead to significant changes in cardiovascular function. While thoracic and lumbar SCI represent 50% of the injuries and 40% the clinical studies which assessed the effects of low-thoracic/lumbar SCI on CVD in humans [47], the effects of low-thoracic/lumbar SCI in pre-clinical models have not been assessed. Additionally, very few studies have assessed the cardiovascular (and metabolic) dysfunctions at chronic time points in animal models. In the present study, we focused on evaluating the impact of thoracic-8 vertebrae (T8) SCI contusions on the cardiometabolic function following long term injury. The aim is to examine the changes in cardiometabolic functions following SCI without altering the sympathetic control of the heart at chronic time points in mice. Male mice received T8 contusions at two severity levels and were monitored by echocardiography, EchoMRI, GTT, body weight, behavior assessments of functional regeneration, and histological evaluation of the liver, heart, and spinal cord after 20 weeks. Results revealed new aspects of cardiometabolic alterations that occur after SCI, revealing the critical role that SCI severity plays on CVD.

## 2. Materials and Methods

### 2.1. Animal Care and Procedure for SCI

All experimental animal procedures were approved by the Texas A&M University Institutional Animal Use and Care Committee (IACUC). C57BL/6 mice were purchased from Taconic and bred in our vivarium. Six-month-old male mice were randomly assigned to one of three treatment groups: sham, moderate (50 kdyn), or severe (75 kdyn) SCI. Each cage housed up to 5 mice, which were housed in a climate-controlled facility in ventilated cages with a 12-h light/dark cycle. All mice were fed a control diet with ad libitum access to water. SCI was induced by contusion to the spinal cord as previously described [53]. Briefly, mice were anesthetized by 3% induction and maintained on 1.5% of isoflurane inhalation and the surgical site at thoracic T8 was shaved and sterilized by isopropyl alcohol. The surgical site was incised, and a partial dorsal laminectomy was performed at T8-9 to expose the spinal cord without penetrating the dura. SCI was induced using the NYU-MASCIS weight-drop impactor [54] at 50 kdyn for moderate and at 75 kdyn for severe SCI (2-sec dwell time). The back muscles were sutured, and the skin was closed with surgical glue. Sham mice received only a laminectomy identical to the other groups without a contusion. After surgery, all mice received saline solution and buprenorphine (0.05 mg/kg, Par Pharmaceutical Chestnut Ridge, Chestnut Ridge, NY, USA) daily for 3 days for hydration and pain, respectively, and penicillin (5 mg/kg/day, Bayer Healthcare LLC, Animal Health Division Shawnee Mission, Shawnee, KS, USA) once daily for 7 days to prevent secondary infection. Mice were monitored daily, with their bladder expressed manually twice every day until the mice were able to urinate without assistance or till the end of the experiment.

### 2.2. Behavioral Assessment

Motor functional recovery of hindlimbs was assessed by the Basso Mouse Scale (BMS) and rotarod tests. BMS test was performed as previously described [55,56]; mice were observed for 5 min by two observers blinded to injury type groups. Many features were noted, including ankle movements, stepping pattern, coordination, paw placement, trunk instability, and tail position, with a minimum score of 0 (no movement) to a maximum score of 9 (normal locomotion). Both observers agreed on each of the final scores for each mouse and the average score of all mice within a group is considered the final score for that group.

Rotarod testing was performed as previously described [55], with one individual blinded to injury type performed the test. Mice are placed on a rod (Ugo Basile, Gemonio, Germany) rotating at increasing speeds from 5 to 50 rpm in 3-min intervals with constant acceleration. The latency to fall (in seconds) was averaged between two trials per session. Mice are first acclimated to the test for two sessions for five days the week before injury, and one additional session one day before injury (baseline). Both BMS and rotarod tests were performed at baseline and then at day 3, 7, 14, 21, 28, and then every two weeks up to 140 days PI.

### 2.3. Body Composition Analysis

All mouse body compositions (fat tissue % and lean tissue %) were determined by an EchoMRI-100 quantitative magnetic resonance whole body composition analyzer (EchoMRI-100H, Echo Medical Systems, Houston, TX) as previously described [57]. Each mouse was weighed and scanned without anesthesia. Mice were scanned at baseline prior to injury and then at 4-, 8-, 12-, 16-, and 20-weeks PI. The body weight of each mouse was collected using a digital scale at 7, 14, 21, and 28 days and then monthly thereon until the end of the experiment.

### 2.4. Liver Ultrasound Image Acquisition and Analysis

Mice liver parenchyma was assessed for changes in structures before and after SCI using ultrasound imaging as described previously [58] with modifications as described below. Briefly, mice were anesthetized with 3% isoflurane and maintained with 1.5% isoflurane. Heart rate and temperature were monitored throughout the procedure. The abdominal cavity was shaved before applying Nair to remove the remaining hair. Ultrasound gel was then placed on the mouse’s abdomen as a final step before the probe was applied to image the liver and kidney. The images were taken using a VisualSonics 3100 high frequency machine along with MX 550 D transducer probe. For consistency, two-dimensional B-mode images were acquired with the following acquisition settings (frequency = 40 MHz, frame rate = 165 fps, gain = 35 dB, depth = 15 mm, width = 14.08 mm, dynamic range = 60 dB). Images captured the entirety of both organs separately. After imaging, mice were allowed to recover in a cage and observed for signs of pain or discomfort. The echogenicity of the liver was examined and analyzed using ImageJ software as previously described [58] with modifications described below. Three regions of interest (ROI) plane were selected manually surrounding the portal vein excluding the hepatic vessels and imaging artifacts (area circle size 1 cm^2^). The mean gray value of each of the three circles was calculated at three different areas for each ROI plane, averaged, and analyzed for each mouse. The same procedure was applied to the kidney images (around the cortex area), except for the circle area size of 0.5 cm^2^, which was used as an internal control. The intensity of liver images was normalized to the kidney images for each mouse. Data are presented as a hepatic to renal (H/R) ratio.

### 2.5. Echocardiography Analysis

Mice were scanned with echocardiogram to assess cardiac structure and function at different time points as previously described [59]. Mice were imaged under anesthesia with isoflurane (induction at 3% and then maintained on 1.5%) with a heart rate of 400 to 500 beats per minutes. Chest hair was removed, and warm ultrasound transmission gel was applied. Parasternal short axis view of the heart with M-mode echocardiograph was acquired using VisualSonics 3100 high frequency machine along with MX 550 D transducer probe with 40 MHz center frequency. Parasternal long axis B-mode ultrasound was used as a reference image for the M-mode acquisition of the short axis. To ensure comparison between all measurements, focus was emphasized on the midventricular level of the heart, by identifying the papillary muscles. Left ventricle measurements were performed using the auto LV analysis tool by tracing the internal diameters of the ventricle, averaged from three consecutive cycles for each animal [59,60,61,62,63]. To limit noise caused by respiratory movements, images were acquired when the mice was not actively breathing, as assessed with the respiratory rate provided with the ultrasound system. The functional parameters and anatomical measurements of LV that were assessed include (left ventricle internal diameter (LVID), left ventricular (LV) mass, left ventricular posterior wall thickness (LVPW), left ventricular anterior wall thickness (LVAW), cardiac output (CO), stroke volume (SV), ejection fraction (EF), and fractional shortening (FS) as previously described [59,60,61].

### 2.6. Glucose Metabolism

Intraperitoneal glucose tolerance test (IPGTT) was performed before SCI surgery and at 14 weeks and 20 weeks PI as previously described [59]. Briefly, after mice fasted for 16 h, a baseline fasting blood glucose level was recorded before each mouse received intraperitoneal administration of 20% (2 g/kg) glucose solution (Sigma, Kawasaki, Japan). Blood glucose levels were recorded 15, 30, 60, and 90 min after administration. Blood glucose levels were quantified using a glucose meter (Bayer Contour Next EZ blood glucose meter, Bayer HealthCare, IN, USA) by taking a blood sample (<5 µL) from a small incision made at the tip of the tail using clean surgical scissors. Area under the curve (AUC) for blood glucose levels in each mouse during IPGTT was calculated using GraphPad Prism (Prism 9.0, GraphPad Software, San Diego, CA, USA).

### 2.7. Triglyceride, Cholesterol, and Insulin Analysis

Liver samples were analyzed for triglyceride content and concentration. Liver medial lobe samples were collected fresh from each mouse right before perfusion. Samples were then immediately flash frozen using liquid nitrogen and stored at −80 °C until testing. Approximately 0.25 g of liver tissue were homogenized in 2:1 chloroform methanol, mixed with 1 mol/L CaCl_2_, and centrifuged for 15 min at 13,000 rpm at 4 °C as previously described [64]. Then, 400 µL of the lower lipid phase was removed and placed in a fume hood to allow for evaporation. Once evaporated, the samples were reconstituted in isopropanol and directly analyzed using Infinity^TM^ Triglycerides Liquid Stable Reagent (Cat. No. TR22421, ThermoFisher Scientific, Waltham, MA, USA) according to the manufacturer’s instructions. The resulting assay values were then normalized to the liver tissue mass to produce absorbed triglyceride content (µg/mg).

Plasma samples were analyzed for triglyceride content, cholesterol, and insulin concentrations. Blood samples were collected from the submandibular vein at 18 weeks post SCI and centrifuged at 14,000 rpm for 15 min at 4 °C to separate the plasma. Plasma triglyceride concentrations (mg/dL) were analyzed using Infinity^TM^ Triglycerides Liquid Stable Reagent according to the manufacturer’s instructions. Free cholesterol concentration (µM) was determined using Invitogen^TM^ Amplex^®^ Red Cholesterol Assay Kit (Cat. No. A12216, ThermoFisher Scientific) per the manufacturer’s instructions. Lastly, plasma insulin concentrations (ng/mL) were assessed using Ultra-Sensitive Mouse Insulin ELISA Kit (Cat. No. 90080, Crystal Chem, Elk Grove Village, IL, USA) according to the manufacturer’s instructions.

### 2.8. Histological Analysis of Heart and Liver Tissues

Mice were perfused. Heart and liver tissues were removed and fixed in 4% paraformaldehyde (PFA) in 1× PBS for 24 h before incubating in 15% then 30% sucrose solution for 24 h each. Heart samples were then cut in the middle across the transverse plane. The lower parts of the hearts (containing the apex) were embedded. Left lobe liver sections were embedded such that the tissue closest to the portal triad would be sectioned. Both heart and liver tissues were placed in OCT compound (Cat. No. 625501-01, Sakura Finetek, Torrance, CA, USA, Inc., Torrance, CA, USA) for embedding. Heart cross sections from the middle point of the heart and left lobe of liver transverse sections were cut at 8 µm thickness with 50 µm between each section using a cryostat (Leica Biosystems CM3050 S). Sections were stained with Masson’s trichrome staining (Sigma-Aldrich, HT15-1KT) according to the manufacturer’s instructions to examine perivascular accumulation of collagen in the tissue. Analysis and quantification of fibrotic areas were performed as previously described [65] where 3–4 sections per tissue were imaged at 20× on a Zeiss Axio Observer 7 fluorescent microscope. Quantification of the left ventricle and liver sections was performed using ImageJ software (NIH, Bethesda, MD, USA). Briefly, the total area for each image was computed and the color-based threshold tool was used to highlight the fibrotic (blue) regions of tissue in each image. To ensure standardization, the hues were set at 130 and 190 for each fibrotic image.

LV thickness was measured using Zen Lite software. For each mouse, 3–4 representative heart sections were chosen. For each section, the LVAW and LVPW thickness was measured by drawing three lines spanning the entire wall thickness and then averaging the length of three drawn lines. The precise locations of these lines were chosen to measure the thickness at the best tissue structural integrity (i.e., no tears resulting from mounting) and avoiding the regions surrounding the papillary muscles.

### 2.9. Tyrosine Hydroxylase (TH) Immunofluorescence Staining of the Heart and Liver

To assess if the T8 contusion SCI affected innervation of the heart and liver, both heart and liver sections were stained for Tyrosine Hydroxylase (TH). Three consecutive transverse heart and left lobe liver sections from each mouse were cut at 10 μm thickness and directly mounted onto Superfrost^®^ Plus MicroSlides (Cat. No. 48311-703, VWR, Radnor, PA, USA). Sections were washed 3 times using 1× PBS and blocked for 1 h using 5% normal horse serum diluted in 0.4% Triton X-100 in 1× PBS. Sections were incubated overnight at room temperature in TH antibody (1:500, Cat. No. AB152, Millipore Sigma, Burlington, MA, USA) diluted with 2% normal horse serum in 0.4% Triton X-100 in 1× PBS. After washing, secondary antibody Alexa Fluor Plus 488 (1:500, Cat. No. A32814, ThermoFisher Scientific) diluted with 2% normal horse serum in 0.4% Triton X-100 in 1× PBS was applied to the sections for 2 h at room temperature. Sections were then incubated in DAPI (1:2000, Cat. No. 62248, ThermoFisher Scientific) and washed 3 times using 1× PBS and cover slipped for examination under the microscope. Images were taken at 20× magnification on a Zeiss Axio Observer 7 fluorescent microscope. Each image was acquired as a z-stack and the maximum projection was used for quantification. For heart sections, 3 images for each region of the LV anterior wall (LVAW), LV posterior wall (LVPW), and LV lateral wall were quantified for TH staining using Quantitative Pathology and Bioimage Analysis software (QuPath v0.3.0, Scotland). In brief, pixel classifiers were programmed to distinguish between positive, negative, and background staining. Total positive and negative areas (µm^2^) were generated and the ratio of the two provided a percentage of TH positive staining within each section. For each mouse, an average TH positive staining was calculated by averaging the percent positive of the LVAW, LVPW, and lateral wall portions for all three heart sections. The average percent positive for the three sections was then averaged to produce the final TH percent positive value for each mouse.

Quantification of TH staining within the liver sections was done using QuPath v0.3.0. Three liver sections were quantified per animal. For each liver section, 3 vessels between 100 μm and 200 μm were chosen for a complete and accurate analysis. A brush tracer tool with a standardized diameter of 26 µm was used to create an annotation around the outer edge of the vasculature. The border region was then filled in to include the entire vessel within the annotation. For each liver section, an average TH positive staining percentage was calculated from the analysis of the three annotated vessels. An average TH positive staining was calculated by averaging the percent positive for all three vessels per sections, followed by averaging the percent positive for the three liver sections to produce the final TH percent positive value for each mouse.

### 2.10. Histological Analysis of SC Injury

To examine the degree of injury caused by the impactor, spinal cords were harvested from contused mice at 20 weeks PI. Mice were anesthetized with intraperitoneal injection of 100 µL of Sodium pentobarbital (Fatal-Plus^®^) before being euthanized via perfusion with 4% paraformaldehyde (PFA) in 1x phosphate buffer solution (PBS) (Cat. No. 14200075, Life Technologies, Carlsbad, CA, USA). Samples were soaked in 15% and 30% sucrose overnight prior to embedding in OCT compound (Cat. No. 625501-01, Sakura Finetek USA, Inc., Torrance, CA, USA), followed by longitudinal sectioning at a thickness of 25 µm using a cryostat (Leica Biosystems CM3050 S). Fixed sections of spinal cord tissue were washed (3× with 0.4% Triton X-100 in 1x PBS), then blocked using 5% normal horse serum (VWR 102643-676, diluted in 0.4% Triton X-100 in 1× PBS) for 1 hour. Sections were incubated with primary anti-glial fibrillary acidic protein (GFAP) antibody (1:500, Cat. No. 13-0300, ThermoFisher Scientific) diluted in 0.4% Triton X-100 in 1× PBS and incubated at 25 °C overnight. The sections were then washed before being incubated with Alexa Fluor Plus 488 secondary antibody (1:1000, Cat. No. A32814, ThermoFisher Scientific) in 0.4% Triton X-100 in 1× PBS for 1 hour and DAPI (1 μg/mL, Cat. No. 62248, ThermoFisher Scientific) for 5 min. After being washed once more, the slices were mounted, and cover slipped (Cat. No. F6182, Sigma) before being imaged at 20× on a Zeiss Axio Observer 7 fluorescent microscope. The contour (polygonal) tool in the Zen 3.2 (Carl Zeiss AG, Jena, Germany) software was used to trace and measure lesion and cavity size. The lesion size was measured by tracing the glial scar border labeled with GFAP surrounding a DAPI+ inner region. Cavity size was measured by measuring the region within the spinal cord without any visible nuclei yet surrounded by GFAP labeled glial scar. Three spinal cord sections per animal were imaged and analyzed.

### 2.11. Blood Cytokines Measurements

Plasma concentrations of IFN-γ, IL-10, IL-17A/CTLA8, IL-6, TNF-α, and IL-1β were measured using MILLIPLEX^®^ Mouse Cytokine/Chemokine Magnetic Bead Panel (MCYTOMAG-70K-Millipore Sigma, Burlington, MA) according to the manufacturer’s instructions. Briefly, approximately 100 µL of blood samples were collected from the submandibular facial vein using a sterile lancet in BD Microtainer^®^ blood collection tubes (cat# BD 365985). Blood samples were collected before SCI surgery and immediately prior to necropsy. Samples were centrifuged at 15,000× *g* for 15 min to separate the plasma. Plasma samples were stored at −80 °C until the assay was conducted. Plasma samples were diluted 2-fold in Assay Buffer provided in the kit per the manufacturer’s recommendation. Samples, quality controls, and standards were aliquoted into the provided 96-well plate in duplicate followed by the antibody-immobilized beads. After a 16-h incubation at 4 °C and respective wash steps, detection antibodies and Streptavidin-Phycoerythrin were added. Washing was conducted using a hand-held magnet. After the completion of the protocol, the plate was analyzed using a Luminex^®^ 200^TM^ (cat# LX200-XPON3.1) multiplex analyzer with the xPONENT 3.1 software. The data represent the average mean fluorescent intensity values from the duplicates. The values at the chronic 5-month timepoint were divided by the baseline values to determine the “fold change from baseline” value plotted.

### 2.12. Statistical Analysis

Two-way ANOVA test was followed by Tukey’s multiple comparison post hoc analysis to assess the differences between the groups. All analyses were performed using GraphPad Prism (Prism 9.0, GraphPad Software, San Diego, CA, USA). Differences were considered significant at *p* ≤ 0.05, tendencies at *p* ≤ 0.10. Data are presented as means ± SEM.

## 3. Results

### 3.1. Experimental Design and Data Collection

To examine the impact of SCI severity after T8 contusion on the cardiometabolic functions, six month old male mice (corresponding to ~25 years old in humans, when the first peak of SCI is observed [66]) were randomly sorted into sham, moderate SCI (50 kdyn), or severe SCI (75 kdyn) groups. There was n = 6 for sham, n = 8 for moderate SCI, and n = 9 for severe SCI at the beginning of the experiment. Mice were tested for parameters of body weight, EchoMRI, echocardiography and liver ultrasound imaging, IPGTT, and BMS and Rotarod scores as described in the experimental shown in Figure 1 and Table 1.

### 3.2. Sustained Reduction of Motor Function after Chronic SCI

Motor functional recovery was assessed following SCI by BMS and Rotarod at multiple timepoints. BMS scores are shown in Figure 2A. Sham mice were not affected by the sham surgical procedure without contusion. For mice that received moderate SCI, the BMS scores dropped to 1.06 ± 0.32 at day 2 after SCI and gradually recovered up to a score of 3.25 ± 0.55 at the end time point (20 weeks). For mice that received severe SCI, BMS scores dropped near to 0.55 ± 0.43 at day 2 after SCI and gradually recovered up to a score of 2.6 ± 1.05 at the end time point. Interestingly, some differences emerged between moderate and severe SCI mice, where severely injured mice showed a significantly decreased recovery timeline compared to moderately injured mice. Rotarod assessment was also performed to determine the extent of hindlimb function on mice (Figure 2B). Mice from both injuries exhibited a significant decrease on time on the rotarod after SCI compared to sham controls. No significant differences were observed between severe and moderate injured mice. There was significant attrition in SCI severe group. Indeed, while all the sham and moderately injured mice survived the 20 weeks timepoint, 45% of the severely injured mice died prior to the study endpoint of 20 weeks (only 5/9 survived, Figure 2C).

### 3.3. Severity-Dependent Reduction of Body Weight and Body Composition

As expected, both moderate and severe groups lost weight within the first week post SCI (Figure 3A). While this loss stabilized in the moderate group, the weight dropped further by week 2 in the severely injured group. Both groups slowly regained some weight over the 20-week period, without reaching their baseline levels. Both groups were significantly lower than the sham group. While not statistically different, the moderate group tend to recover weight better than the severe group. Fat and lean body composition were measured using EchoMRI. By 4 weeks PI, the percentages of body fat and body lean were significantly reduced in the severe group compared to the sham group (Figure 3B,C). Similarly, the moderate group presented a significantly reduced percentage of body fat compared to sham (Figure 3B), while the percentage of lean body mass was also reduced without reaching statistical difference (Figure 3C).

### 3.4. Severe SCI Induced Stronger Liver Pathology

Liver ultrasound analysis was performed prior to SCI and at 4-, 8-, 12-, 16-, and 20-weeks PI to examine the liver shape and structure (Figure 4A). By 12 weeks PI, the severe group tended to be increased in the liver echogenicity, although not reaching statistical levels compared to the sham and moderate injured groups (Figure 4B). Histological analysis of the liver at 20 weeks exhibited a significant increase in fibrotic tissues in severely injured mice and a non-significant upward increase in moderate group, compared to the sham animals (Figure 4C,D).

### 3.5. Severe SCI Induced Cardiac Dysfunction

Echocardiography was performed to assess cardiac structure and function following SCI. In the severe SCI group, LVID during systole (LVID;s) was reduced at 16 weeks PI and reached statistical significance at 20 weeks PI (Figure 5A). No significant reduction in LVID during diastole (LVID;d) was observed (Figure 5B). The reduction of LVID;s was associated with the reduction in LV volume (Figure 5C) and no significant difference in LV volume was observed in LVID;d (Figure 5D). Interestingly, both ejection fractioning (EF) and fractional shortening (FS) trended upward at 16 weeks and significantly increased at 20 weeks PI in the severely injured group compared to the sham and moderate group (Figure 5E,F, respectively). Masson’s trichrome staining of the left ventricular sections showed significant change in collagen accumulation in severe SCI and an upward trend in the moderate group compared to their sham controls (Figure 6A,B). While there were no significant differences in other cardiac parameters such as LV mass, SV, CO, HR, and LVPW and LVAW thickness using echocardiography scanning (Table 2), histological analysis of the LVPW and LVAW thicknesses revealed a significant increase in LVPW and LVAW thickness compared to moderate and sham groups, respectively (Figure 6C).

### 3.6. No Changes in Glucose or Lipids Metabolism after Chronic SCI in Mice

Using IPGTT, we did not observe any significant changes in glucose metabolism up to 20 weeks PI. This is consistent with previous reports which state that SCI induces slight change in serum glucose at 23 days PI [67]. IPGTTs were performed prior to injury, then at 14- and 20-weeks PI. Results showed that the glucose concentrations and area under the curve (AUC) were similar between the groups pre-injury (Figure 7A). No significant differences were observed in glucose concentration and AUC at 14- and 20-weeks PI between any of the groups (Figure 7B,C). Fasting glucose concentrations were also similar between all groups up to 20 weeks PI (Figure 7D).

Complementary to the lack of a significant difference in glucose metabolism, no significant changes in plasma insulin levels between the sham, moderate, or severe SCI models were found at 18 weeks PI (Figure 7E). The 18 weeks PI results show that there was also no significant difference in plasma cholesterol concentrations (Figure 7E). Although there was no statistically significant change in triglyceride levels both in the plasma (Figure 7E) and the liver (Figure 7F), there was a trend of decrease in the severely injured model (*p* = 0.13 between moderate and severe SCI for plasma TG levels). A decrease in triglyceride levels coincided with our EchoMRI findings showing a much lower compositional fat % in severely injured mice. Of note, no difference in temperature was observed at 20 weeks in between the groups (not shown).

### 3.7. Severe SCI Induced Changes in SC Injury

The lesion and cavity size at the spinal cord injury sites were analyzed to compare the size of the injury following different severities (Figure 8A,B). There is an upward trend in the lesion size *p* = 0.06 (Figure 8C) and cavity size *p* = 0.2 (Figure 8D) and a significant increase in total injury size *p* = 0.02 with increasing severity. The severe group is 40% more likely to have a cavity at the injury site relative to the moderate group. There is a significant negative correlation between total injury size and BMS score *p* = 0.04 (Figure 8E) and a non-significant negative correlation between total injury size and Rotarod performance *p* = 0.06 (Figure 8F). Interestingly, we observed a trend for a positive correlation between total injury size and LVAW;s, not reaching statistical difference *p* = 0.09 (Figure 8G), but suggesting that mice with the most severe injuries might develop more cardiac dysfunctions.

### 3.8. SCI and Plasma Cytokines

To assess the effects of chronic SCI on cytokine expression, the relative plasma concentrations of IFN-γ, IL-10, IL-6, TNF-α, and IL-1β were measured. We observed an increase in IL-1β concentration in the severe SCI group *p* = 0.05 (Figure 9C). No significant differences were observed between any cohort for the other cytokines, although there was a trend for a decrease in IL-10 (Figure 9B) and an increase in IL-6 (Figure 9A) in the severely injured mice.

### 3.9. SCI, Heart, and Liver Innervation

To assess how SCI could potentially change innervation of both the heart and the liver, the percent of tyrosine hydrolase (TH) stain was measured in each tissue (Figure 10A,C). Between the sham and severe injury models for the heart, no significant difference in positive TH percentage was found (Figure 10B). Likewise, there was no significant difference in positive TH percentage in the liver (Figure 10D).

## 4. Discussion

The main findings of this study are that chronic SCI at T8 level: (1) increases the liver echogenicity, which is associated with increased liver fibrosis; (2) reduces the LVID and increases the FS that are found to be associated with increased cardiac fibrosis and LV thickness indicating severe hypertrophy; (3) induces the sustained loss of motor functions; and (4) leads to the reduction in body weight, lean and fat percentage. Importantly, these perturbations in cardiometabolic functions vary significantly depending upon the severity of the injury and play a key role in determining the long-term health outcome following SCI.

### 4.1. SCI Severity and Hepatic Dysfunctions

The liver plays an essential role in metabolic function and SCI is associated with liver abnormalities in humans and rodents [37,39]. We observed a non-significant increase in liver echogenicity at 12 and 20 weeks after severe injury. Several factors may lead to changes in liver echogenicity including liver steatosis [68]. Changes in the hepatic echogenicity may be induced by the infiltration of pro-inflammatory cytokines and chemokines in the liver due to the trauma [40]. Interestingly, these changes in echogenicity were associated with an increase in fibrotic tissues in relation to the injury severity. In humans, ultrasound imaging is not routinely performed to assess liver dysfunction in people with chronic SCI [37]. Our data suggest that changes in echogenicity obtained from ultrasound imaging after chronic SCI, even if not significant, are correlated with significant increase in histological fibrotic changes of the liver. Therefore, because of the risks associated with a liver biopsy to assess for hepatic diseases, noninvasive examination of liver echogenicity following SCI may be of high interest to monitor the development of hepatic dysfunction in humans. This could be performed in addition to testing for serum markers such as alanine aminotransferase and aspartate transaminase (AST) [67].

Several studies have reported that SCI patients developed metabolic dysfunctions such as insulin resistance and impaired glucose tolerance [20,21,69,70]. Using IPGTT to assess glucose tolerance, our results revealed no differences between the groups at any time points up to 20 weeks after SCI. These results are consistent with a prior study reporting that glucose intolerance was not observed in SCI male mice at 56 or 84 days after T10 SCI transection [71]. However, the same study described an elevated fasting blood glucose in these mice. Another group reported that moderate T8 contusion in female rats led to no significant change in glucose levels at 23-days post SCI [67]. Importantly, female rats with a complete T3 transection showed a significant enhancement in glucose handling at 16 weeks post SCI, with lower serum insulin concentrations [72]. Our data do not show a change in insulin level at 18 weeks post SCI. However, these measurements of insulin levels were not performed on fasted animals. Future work is needed to better characterize the complex insulin/glucose relationship in mice models of SCI. Altogether, this suggests that the injury severity (contusion vs. transection), the level of the injury (high or low thoracic), and the timing post-SCI are important factors impacting metabolic functions [69]. Indeed, both the hepatic infiltration of fatty acids and/or inflammation following SCI and the disruption of the hepatic nervous system may impact hepatic functions. Our data suggest that severe T8 contusion does not alter hepatic TH innervation. However, it is of high interest to better understand how the level and severity of the injury may play a key role in directly or indirectly modulating the hepatic nervous system and the hepatic functions.

### 4.2. SCI Severity and Cardiac Functions

The development of cardiovascular dysfunction following SCI is one of the leading causes of death among SCI patients [73]. Cardiovascular dysfunction after SCI is dependent on the level and the degree of the injury [74]. Disruption of the cardiovascular nervous control by SCI leads to autonomic dysreflexia especially if the injury level is at T6 or above [46]. However, a T8 contusion model does not disrupt the cardiovascular sympathetic control, suggesting that the cardiac changes we have observed are due to other factors. Our data suggest no significant changes in TH innervation in the LV after chronic SCI. One could speculate that the reduction in survival observed in severe SCI (only 55% at 5-month post SCI) is the result of these cardiovascular complications. We could not confirm this hypothesis as we were not able collect the tissue from the animals found dead. Our results indicated that, during systole, both the LVID;s and volume were reduced in severe SCI at 16 weeks, and significantly at 20 weeks PI. This was correlated histologically with the increase in the left ventricle thickness. It is known that SCI individuals have smaller LV volumes and mass [75]. In rat models of SCI, West et al. reported a reduction in LV dimensions at 6 weeks post T3 SCI [76] and cardiac atrophy, reduced myocardium contractility, and increased fibrosis [77].

Our data showed an increase in the ejection fraction (EF) at 20 weeks PI in the severely injured group compared to the sham and moderate group. In humans, an EF >75% is a sign of hypertrophy cardiomyopathy. However, previous studies reported no differences in EF between SCI and able-bodied individuals despite LV volumes reduction [75]. In our data, EF is 58.1% in sham, 62.3% in moderate, and 70.4% in severe SCI. These severely injured mice also developed LV thickness and hypertrophy shown by histological analysis. While echocardiography scanning showed no significant differences in other cardiac parameters such as LV mass, SV, CO, HR, and LVPW, and LVAW thickness (Table 2), histological analysis revealed a significant change in LVPW, and LVAW thickness induced by severe SCI. Squair et al. observed a reduction in LVID;s, LVID;d, Volume;s and Volume;d [51]. They also reported a non-significant increase in EF in T3 severe contusion model in rat. However, we must acknowledge the limitation of this type of echo measurements and that short axis view analyses can infer volumetric measurements, and impact the EF calculated [78]. EF is calculated using the LV chamber volume during end systole (Volume;s) and end diastole (Volume;d) with this equation [EF = (Volume;d–Volume;s)/ Volume;d × 100]. Our hypothesis is that because of the LV hypertrophy in the severe SCI group, as measured by histology, the left ventricle contains a smaller amount of blood (reduced Volume;d). In this situation, with a Volume;s not decreasing proportionally, an increase in EF would be measured (as in our data). Because of this and the increase in perivascular fibrosis in the heart, we believe the severe SCI group present cardiac dysfunctions. Future experiments using blood pressure measurement, telemetric devices, or echocardiography using doppler analysis and B-mode/long-axis measurement will allow for the confirmation of these observations.

Another critical aspect of cardiac remodeling is the increase in collagen, often synonymous with fibrosis, in the LV tissue of both SCI groups, with more accumulation in the severity SCI. Outside of a SCI, myocardial fibrosis can be triggered by several factors including mechanical forces, inflammation [79], neurohormonal such as aldosterone [80,81], and others reviewed extensively here [82,83]. LV fibrosis can impair cardiac contractility function, reduce LV chamber size, lead to hypertrophic cardiomyopathy [84,85], LV dysfunction and heart failure [86]. One can speculate that even in moderate SCI, mice may develop hypertrophic cardiomyopathy at later chronic timepoints. The molecular mechanisms underlining these cardiac changes are not well understood. However, chronic inflammation is associated with metabolic syndrome phenotypes, which are causes of cardiovascular diseases. Therefore, sustained inflammatory cytokine activity after SCI may be a trigger for cardiac tissue damage and dysfunction. The direct impact of inflammation on cardiac alterations remains to be determined. Reduction of the systemic inflammation could be a potential target to reduce cardiac complications associated with chronic SCI.

LV thickening may be caused by several mechanisms, including increase in hypertension, diabetes, and aortic valve stenosis. We could not measure the potential changes in blood pressure over time and we did not observe changes in glucose levels. However, we observed significant increase in fibrosis in liver and LV. Therefore, it is possible that fibrosis occurs in other parts of heart, including on the aortic valve, leading to stenosis at chronic time points, and LV hypertrophy. Another possibility is the direct role of systemic inflammation on cardiac functions. Indeed, chronic low-grade inflammation leads to cardiomyopathies, including cardiomyocytes hypertrophy and dysfunctions [87,88,89]. Future work is necessary to understand the mechanism of LV thickening in this chronic T8 severe contusion model in mice, including the changes in cardiomyocytes size and numbers [51].

While we do not observe differences in TH staining in the heart and liver, we have to acknowledge that the T8 contusion model used here can impact innervation in other organs [45,46]. Therefore, any change in blood regulation in these organs induced by the dysregulation of the nervous system may overtime directly impact heart functions and induce hypertrophy and fibrosis through change in blood pressure. Additionally, more work is needed to understand how mid-thoracic SCI could precisely alter heart functions, despite the non-significant change in TH. Indeed, neural remodeling in the autonomic nervous system, and the balance between the sympathetic and parasympathetic systems, may be altered and induce further changes in cardiac functions and regulation. Interestingly, the autonomic nervous system is involved in detecting and modulating inflammation [90]. Therefore, change in the sympathetic tone can not only modulate heart function but also directly alter inflammation. Whether inflammation alters the sympathetic first after SCI of the reversed needs to be determined. 

### 4.3. SCI Severity and Body Composition

Due to the severe trauma induced by SCI which increases the metabolic demand, significant changes in the body weight and body fat may occur [70]. In humans, reports have indicated that people with SCI exhibit a significant increase in body mass index (BMI) [91]. These changes may increase the body fat and reduce lean mass [7,24,92]. Our results showed a reduction in body weight and body fat and lean percentage in both SCI severity models, compared to the sham control group. Severely injured mice lost more body fat and lean mass than moderately injured mice. This prolonged reduction in body weight including fat mass was also reported in rats after 16 weeks of T3 SCI [72]. In this study, the authors related this reduction to possible permanent changes in gastrointestinal transit and absorption and not due to hypophagia. Interestingly, thoracic SCI at level T10 in rats maintained on a low-fat diet did not gain weight compared to SCI rats maintained on high fat diet after 12 weeks post injury, indicating that diet also plays an important role in changes in body composition [93]. Our results showed a reduction in fat and lean percentages compared to sham. This reduction in fat is correlated with the overall reduction in the level of triglycerides in the liver and plasma levels in severe SCI, although it is not significant. Previous work reported some pathological changes to lipid species early on following injury [39,94]. However, our data suggest that when examined 5 months after SCI, the lipid levels appear to be normal with a moderate SCI, and slightly reduced after a severe SCI. Potential increases or decreases within the lipid profile of SCI patients have been shown to be dependent upon numerous factors such as the level and severity of injury, age/sex, and time after injury [38,95,96]. Damage to the liver, notably fibrosis, does remain at this chronic timepoint, suggesting that the liver may be more prone to inflammation or differential response after high fat diet feeding. The lack of fat gain in rodent and the reduction in lipid (hepatic and circulating) is intriguing. Different types of diet, lower food intake, and changes in metabolism may explain the slower fat accumulation following SCI in mice. In this study, we did not monitor dietary intake, which would be recommended for future work. It will also be interesting to challenge this new metabolic state by shocking the system via a large uptake in fat. It would be interesting to see if the mice gain weight (and fat) at all at more chronic time points as aged mice do.

### 4.4. SCI Severity, Motor Function, and Injury Size

Regarding behavioral assessments, BMS testing indicated that, for both SCI severity models, mice did not fully recover their locomotor function at 20 weeks after SCI. Moderate SCI mice recovered better than severely injured mice. Functional recovery in mice is severity-dependent and is not complete in a moderate or severe contusion model of SCI, with animals reaching the maximum of the recovery several weeks after injury. Four months old mice injured at T8-10 (75 kdyn) contusion did not recover fully after 6 weeks post injury [97,98]. Our results assessed recovery at chronic time point (up to 5 months post-SCI) and demonstrate that functional recovery is severity dependent and that injured mice do not further recover over a long period of time.

Our results showed a significant increase in the spinal cord lesion size with the model of injury severity. It has been previously shown that there is a positive relationship between lesion severity and functional outcomes where increase in lesion size may lead to decreased outcomes [99]. Interestingly, liver dysfunction and inflammation at the time of SCI increases the spinal cord lesion size and worsens motor recovery [67]. The changes induced by SCI are affecting several organs and are severity dependent. This is apparent in individuals living with SCI because they are more prone to infections and cardiometabolic disorder [100]. Chronic exposure to increased levels of circulating inflammatory cytokines lead to liver dysfunction and metabolic disease [94]. We observed a significant increase in IL-1β and a trend for upregulation of IL-6 plasma concentrations in the severe group compared to the sham groups. Previous finding showed that IL-1β gene expression was upregulated in the hepatic tissue for at least 21 days post-injury [39]. Furthermore, IL-6 has been implicated in regulations of metabolic function induction of the hepatic acute phase proteins [101]. Hepatic IL-6 expression is increased in animal models of nonalcoholic fatty liver disease (NAFLD), which results in insulin resistance in mice [102]. Considering the role of the liver as a key organ in regulating metabolism, modulating the inflammatory response may reduce liver dysfunction, which in turn may reduce the onset of the severity of cardiometabolic dysfunctions.

## 5. Conclusions

In conclusion, our data suggest that mice with thoracic T8 injury develop numerous hepatic and cardiometabolic alterations over time. Further work will determine if changes in innervation in several organs after SCI, including inflammatory organs such as the spleen and the gastro-intestinal tract, can directly or indirectly influence these health issues. Importantly, it will be of high interest to determine if targeting inflammation will reduce these metabolic outcomes, as well as reduce the morbidity and mortality of people living with SCI.

## Figures and Tables

**Figure 1 biology-11-00495-f001:**
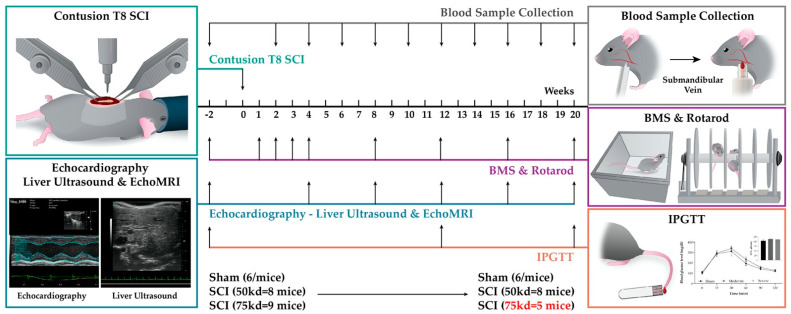
**Schematic diagram of the experimental design.** Six-month-old male mice were used to measure a baseline for Echo MRI, echocardiography (Echo), liver ultrasound, intraperitoneal glucose tolerance test (IPGTT), and body weight (BW). At day 0, T8 contusion SCI was induced, and the same measurements were performed at 4, 8-, 12-, 16-, and 20-weeks post-injury. IPGTT was performed only at the 12 and 20-week time point. Blood plasma was collected twice monthly. For behavioral assessments, BMS and rotarod tests were performed pre-injury and again at days 2 then week 1, 2, 3, 4, 6, 8, 10, 12, 14, 16, 18, and 20 post-injury. At week 20 post-SCI, animals were sacrificed, blood plasma was collected, and tissues (hearts, livers, and spinal cords) were harvested for histological analyses.

**Figure 2 biology-11-00495-f002:**
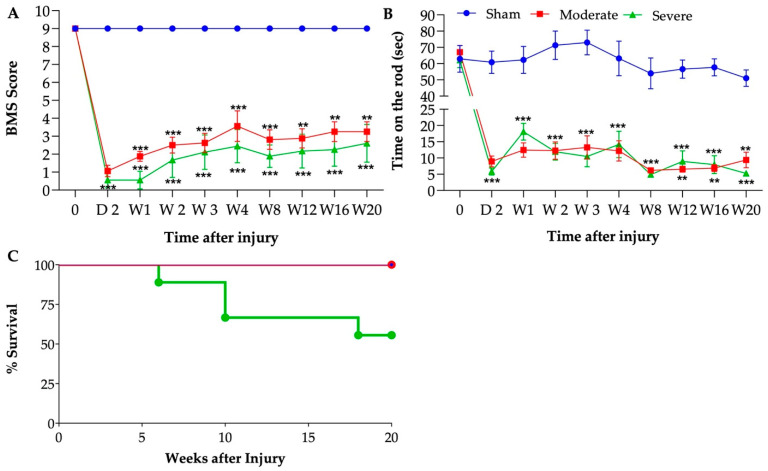
**Behavioral testing after SCI**. BMS scores (**A**), Rotarod score (**B**). Behavioral tests performed at baseline before injury and on days 2, 7, 14, 21, 28, and then monthly PI. (**C**) Percent survival during the experiment. Repeated measures two-way ANOVA: Tukey’s multiple comparisons test was used to determine the differences between the groups. Data presented as means ± SEM, n = 5–9 per group. * *p* ≤ 0.03, ** *p* ≤ 0.002, *** *p* ≤ 0.001.

**Figure 3 biology-11-00495-f003:**
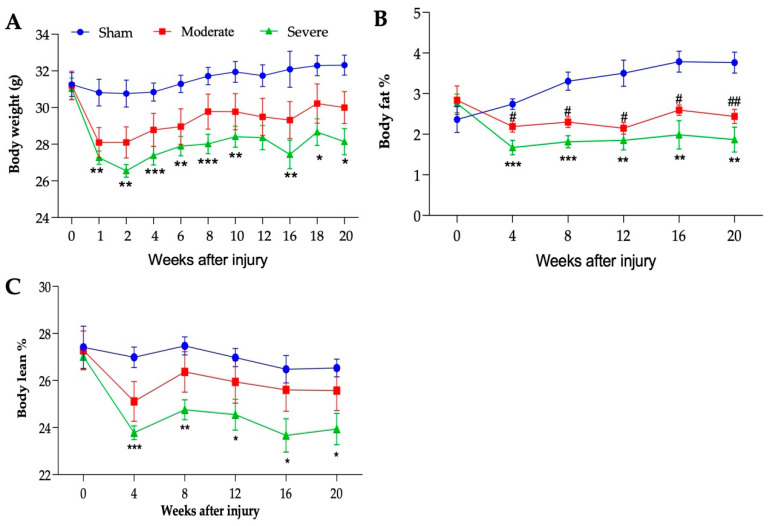
**Injury-severity dependent body composition changes.** (**A**) Body weight (g), (**B**) percentage of body fat, and (**C**) percentage of lean body mass. Statistical analysis was performed using repeated measures two-way ANOVA; Tukey’s multiple comparisons test was used to determine the differences between the groups. Data presented as mean ± SEM, n = 6 (sham), 8 (moderate), and 5–9 (severe). * *p* ≤ 0.03, ** *p* ≤ 0.002, *** *p* ≤ 0.001 difference between sham and severe groups and # *p* ≤ 0.03, ## *p* ≤ 0.002 difference between sham and moderate groups.

**Figure 4 biology-11-00495-f004:**
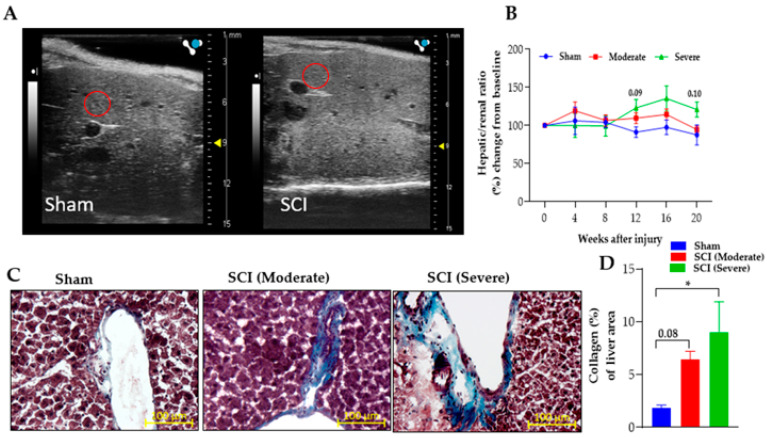
**Injury-severity dependent reduction in liver function.** SCI induced liver intensity and fibrosis. (**A**) Representative images of liver show echogenicity of the liver parenchyma increased in the severe group compared to sham group using ultrasound imaging at 20 weeks PI of sham and severe SCI. Red circles located on the liver images represent areas of interest that were quantified. (**B**) Severe SCI increased liver intensity after 12 weeks PI compared to sham group. Kidney tissue was used as the internal control with data represented as a ratio of hepatic/renal percent change from baseline measurements. (**C**) Representative liver sections stained with Masson’s trichrome of sham, moderate-SCI, and severe-SCI at 20 weeks PI. (**D**) Quantification of collagen contents (blue) in the liver. SCI significantly increased fibrotic tissue in severe injured group and there is a trend towards an increase in the moderate group compared to the sham control. Scale bar = 100μm. Values presented as mean ± S.E.M of 3–4 sections/mouse; n = 6 (sham), 8 (moderate), and 5–9 (severe). Data analyzed by two-way ANOVA. Tukey’s multiple comparisons test was used to determine the differences between the groups; one-way ANOVA for D. * *p* ≤ 0.03, ** *p* ≤ 0.002, *** *p* ≤ 0.001.

**Figure 5 biology-11-00495-f005:**
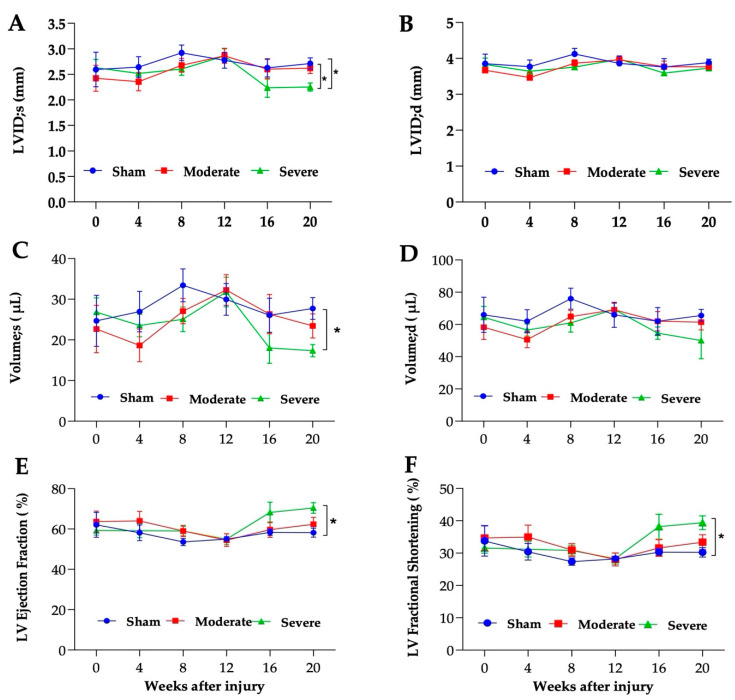
**Injury-severity dependent development of cardiac dysfunction**. Echocardiography assessment. (**A**) Left ventricular internal diameter during systole (LVID;s) and (**B**) during diastole (LVID;d). (**C**) LV volume during systole and (**D**) LV volume during diastole. (**E**) Ejection fraction (EF) and (**F**) fractional shortening (FS). n = 6 (sham), 8 (moderate), and 5–9 (severe). Data analyzed by two-way ANOVA. Tukey’s multiple comparisons test was used to determine the differences between the groups. Data presented as mean ± SEM, * *p* ≤ 0.03, ** *p* ≤ 0.002, *** *p* ≤ 0.001.

**Figure 6 biology-11-00495-f006:**
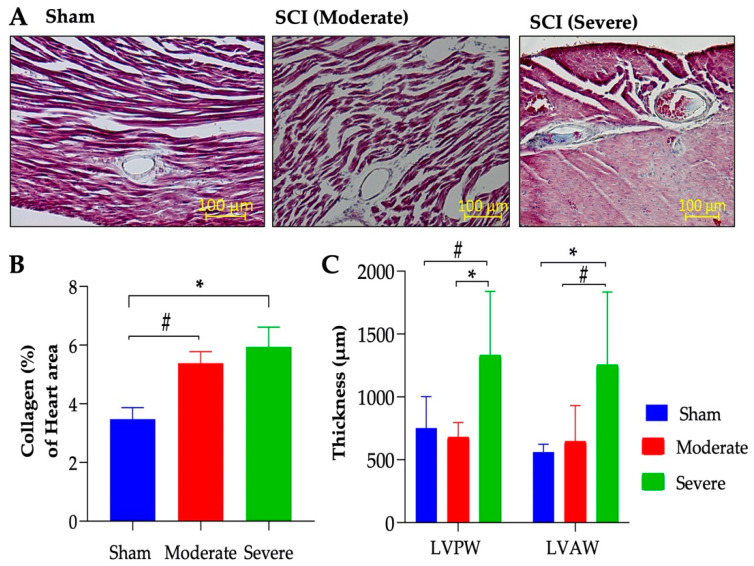
**Injury-severity dependent increase in cardiac remodeling.** (**A**) Representative 40× sections of Masson’s trichrome of sham, moderate-SCI, and severe-SCI after 20 weeks PI. (**B**) Quantification of collagen contents (blue) in the LV tissue. SCI significantly increased fibrotic tissue in the severe injured group and there is a trend to increase in the moderate group compared to the sham control. (**C**). LVPW and LVAW thickness. Scale bar = 100 μm. Values presented as mean ± S.E.M of 3–4 sections/mouse; n = 6 (sham), 8 (moderate), and 5–9 (severe). Data analyzed by ANOVA. * *p* < 0.05 and # *p* < 0.1.

**Figure 7 biology-11-00495-f007:**
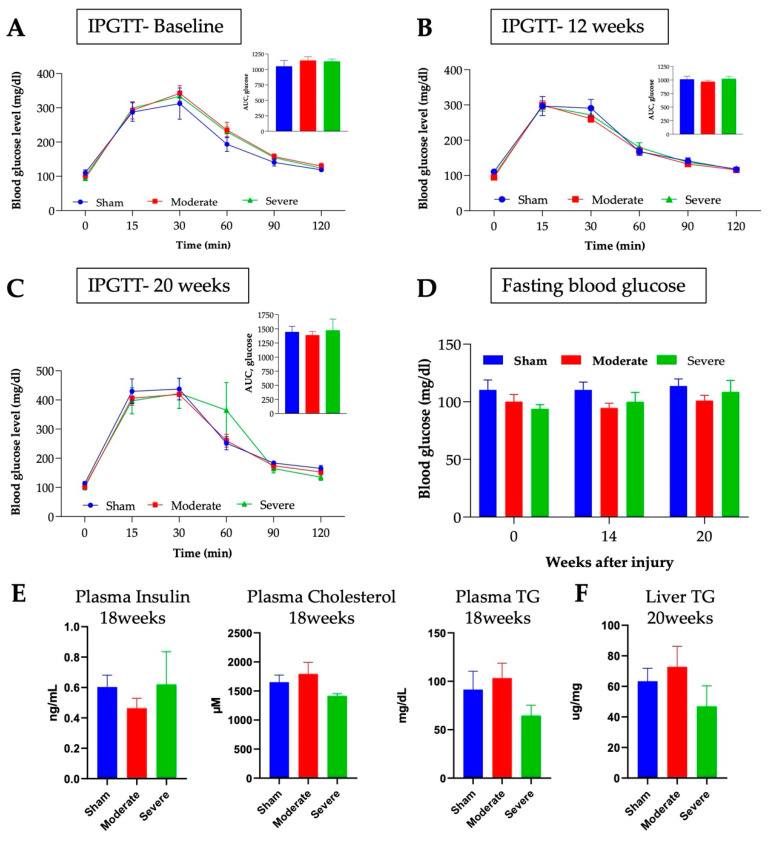
**No changes in glucose metabolism after SCI regardless of the injury severity.** IPGTT was performed pre-injury (**A**), at 12 weeks (**B**), and at 20 weeks PI (**C**). (**D**) Fasting blood glucose concentrations at different time points. No differences are observed for any of these measures. Plasma insulin, cholesterol, and triglyceride levels (**E**) were tested at 18 weeks PI. Liver triglyceride content (**F**) was assessed after euthanasia at 20 weeks PI. Values presented as mean ± S.E.M. N = 6 (sham), 8 (moderate), and 5–9 (severe). Data analyzed by two-way ANOVA or one-way ANOVA (**D**–**F**). * *p* < 0.05.

**Figure 8 biology-11-00495-f008:**
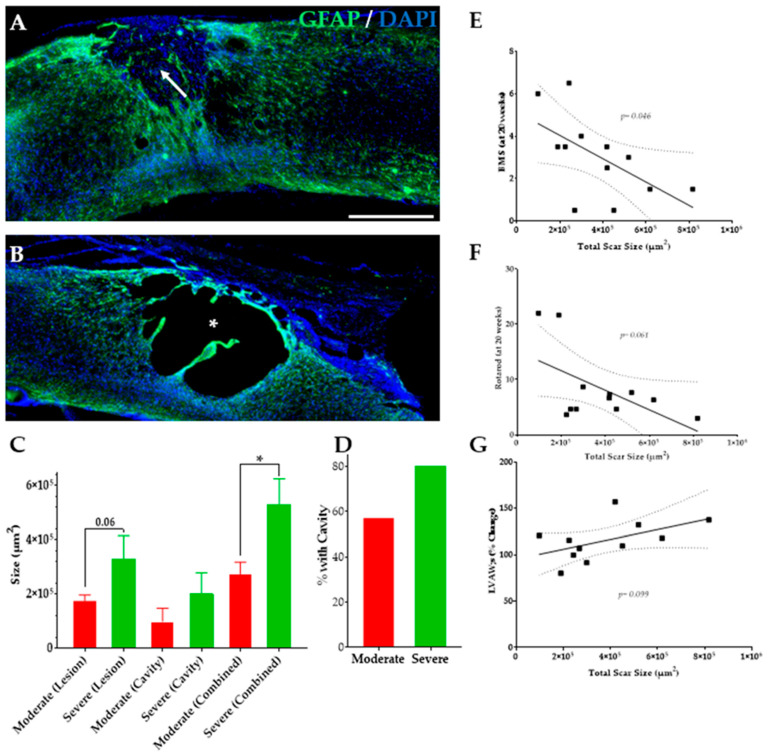
**Lesion size and relation to behavior and changes in cardiovascular measures.** Sample image of spinal cord (severe group) stained with GFAP (green) and DAPI (blue) showing (**A**) a lesion (white arrow) and (**B**) a cavity (white asterisk) site. (**C**) The average lesion, cavity, and total injury size at 5 months post SCI. (**D**) The percent of mice with cavities at the site of injury. Linear trends of the (**E**) BMS score at 20 weeks PI, (**F**) rotarod performance (time on the rod in seconds) at 20 weeks PI, and (**G**) percent change in LVAW;s in relation to the total injury size of moderate and severe groups. Student’s T-test comparing the means of 2 groups. Linear regression analysis to determine correlation between variables. Data presented as means ± SEM, n = 8 (moderate), and 5 (severe). Linear regression trend line ± 95% confidence interval. Scale bar = 500 µm. * *p* < 0.05.

**Figure 9 biology-11-00495-f009:**
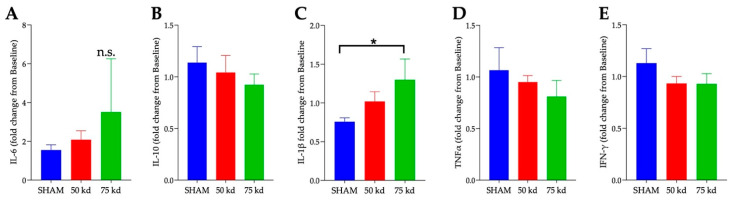
**Injury-severity dependent increase in pro-inflammatory molecule.** Plasma cytokines of (**A**) IL-6, (**B**) IL-10, (**C**) IL-1β, (**D**) TNF-α, and (**E**) IFN-γ at 5 months post injury were quantified using MILLIPLEX assay. Values presented as mean ± S.E.M. n = 6 (sham), 8 (moderate), and 5 (severe). Data analyzed by one-way ANOVA. * *p* < 0.05.

**Figure 10 biology-11-00495-f010:**
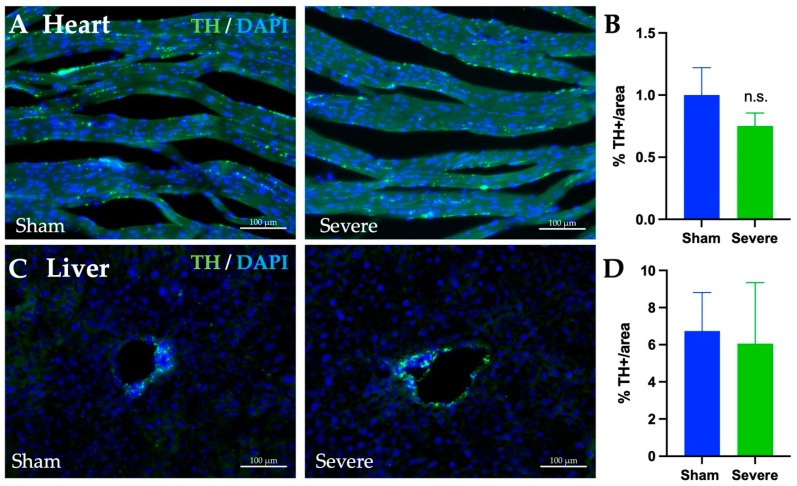
**TH staining in the LV and liver.** Heart (LVAW, **A**) and liver (**C**) 10 μm sections stained for tyrosine hydroxylase (TH) (green) and DAPI (blue) in both sham and sever SCI models. Heart sections from both sham and severe SCI were analyzed for % area of positive TH stain (**B**). No significant differences in % positive was seen. Liver sections from both sham and severe SCI were also analyzed for % area of positive TH stain (**D**). Again, no significant differences in % positive was seen. Values presented as mean ± S.E.M. n = 6 (sham) and 5 (severe). Data analyzed by Student’s T-test comparing the means of 2 groups. * *p* < 0.05. Scale bars = 100 µm.

**Table 1 biology-11-00495-t001:** Summary of number of treatment groups; the experiments were performed up to 20 weeks post SCI.

	Number of Mice			Set of Experiments				
Mice Group	Sham	50 kd	75 kd	BMS & Rotarod	Echocardiography & Liver Ultrasound	EchoMRI	Blood Samples	IPGTT
**Pre SCI**	6	8	9	+	+	+	+	+
**W1**	6	8	9	+	-	-	-	-
**W2**	6	8	9	+	-	-	+	-
**W3**	6	8	9	+	-	-	-	-
**W4**	6	8	9	+	+	+	+	+
**W5**	6	8	9	-	-	-	-	-
**W6**	6	8	9	-	-	-	+	-
**W7**	6	8	9	-	-	-	-	-
**W8**	6	8	8	+	+	+	+	+
**W9**	6	8	8	-	-	-	-	-
**W10**	6	8	6	-	-	-	+	-
**W11**	6	8	6	-	-	-	-	-
**W12**	6	8	6	+	+	+	+	+
**W13**	6	8	6	-	-	-	-	-
**W14**	6	8	6	-	-	-	+	-
**W15**	6	8	6	-	-	-	-	-
**W16**	6	8	6	+	+	+	+	+
**W17**	6	8	6	-	-	-	-	-
**W18**	6	8	5	-	-	-	+	-
**W19**	6	8	5	-	-	-	-	-
**W20**	6	8	5	+	+	+	+	+

**+: Yes, -: No.**

**Table 2 biology-11-00495-t002:** **Parameters of echocardiography results of sham, moderate, and severe SCI at different time points**. Data are means ± S.E.M; n = 5–9 per group. Parameters of cardiac structure and function in mice before SCI and at 4-, 8-, 12-, 16-, and 20-weeks post-SCI of sham, moderate (50-kdyn), and severe (75-kdyn) mice. HR, heart rate; SV, stroke volume; CO, cardiac output; LV Mass, left ventricular mass; LVAW;s, LV anterior wall thickness at the end of systole; LVAW;d, LV anterior wall thickness at the end of diastole; LVPW;s, LV posterior wall thickness at the end of systole; LVPW;d, LV posterior wall thickness at the end of diastole; LVID;d, LV anterior diameter at the end of diastole.

Parameters	Surgery	HR (BPM)	SV (µL)	CO (mL/min)	LV Mass (mg)	LVAW;s (mm)	LVAW;d (mm)	LVPW;s (mm)	LVPW;d (mm)
Time									
**Baseline**	Sham	494 ± 31.8	38.6 ± 3.1	18.8 ± 1.2	128 ± 13.2	1.5 ± 0.1	1.0 ± 0.1	1.4 ± 0.1	1.1 ± 0.1
	Moderate	449 ± 14.7	35.5 ± 2.8	15.8 ± 1.1	134 ± 16.6	1.4 ± 0.1	0.9 ± 0.5	1.7 ± 0.1	1.3 ± 0.2
	Severe	447 ± 15.2	38.1 ± 3.2	17.1 ± 1.6	117 ± 4.8	1.3 ± 0.0	0.94 ± 0.0	1.4 ± 0.1	1.0 ± 0.1
**4weeks**	Sham	493 ± 21.8	35 ± 3.2	17.1 ± 1.3	119.4. ± 5.5	1.3 ± 0.1	0.94 ± 0.0	1.46 ± 0.1	1.1 ± 0.1
	Moderate	482 ± 12.5	32 ± 3.3	15.5 ± 1.7	121 ± 10.2	1.4 ± 0.1	0.97 ± 0.0	1.6 ± 0.1	1.2 ± 0.1
	Severe	497 ± 12.4	32.9 ± 2.1	16.4 ± 1.3	134 ± 17.8	1.45 ± 0.1	1.1 ± 0.1	1.56 ± 0.1	1.1 ± 0.1
**8 weeks**	Sham	451 ± 21.1	40.2 ± 2.8	17.9 ± 0.7	103.3 ± 5.9	1.25 ± 0.1	0.9 ± 0.01	1.1 ± 0.0	0.8 ± 0.0
	Moderate	478 ± 20.1	37.7 ± 1.1	17.9 ± 0.6	119.2 ± 12.1	1.5 ± 0.1	1.0 ± 0.1	1.3 ± 0.1	0.87 ± 0.1
	Severe	472 ± 16.2	35.8 ± 3.2	16.9 ± 1.7	113.3 ± 18.8	1.4 ± 0.1	0.9 ± 0.0	1.4 ± 0.1	0.98 ± 0.1
**12 weeks**	Sham	498 ± 16.3	36 ± 3.9	17.8 ± 1.8	143 ± 16.6	1.4 ± 0.1	1.0 ± 0.1	1.5 ± 0.1	1.16 ± 0.2
	Moderate	492 ± 17.7	36.7 ± 1.5	18 ± 1.0	110.4 ± 7.5	1.3 ± 0.5	0.9 ± 0.0	1.2 ± 0.1	0.89 ± 0.1
	Severe	519 ± 15.7	37..5 ± 0.5	19.4 ± 0.5	110.6 ± 7.6	1.3 ± 0.1	0.9 ± 0.0	1.25 ± 0.1	0.89 ± 0.1
**16 weeks**	Sham	501 ± 24.7	35.9 ± 4.5	17.9 ± 2.3	118.7 ± 9.1	1.3 ± 0.1	0.9 ± 0.0	1.5 ± 0.1	1.1 ± 0.1
	Moderate	476 ± 17.5	35.6 ± 2.0	16.9 ± 1.1	116.2 ± 7.8	1.4 ± 0.1	1.0 ± 0.1	1.3 ± 0.0	0.96 ± 0.0
	Severe	503 ± 3.5	36.5 ± 2.6	18.4 ± 1.3	131.6 ± 20.4	1.4 ± 0.1	0.96 ± 0.0	1.8 ± 0.2	1.2 ± 0.2
**20 weeks**	Sham	449 ± 10	37.8 ± 1.7	17 ± 0.8	142.4 ± 22.2	1.5 ± 0.0	1.1 ± 0.1	1.4 ± 0.1	1.1 ± 0.2
	Moderate	480 ± 6.7	37.9 ± 3.3	18.1 ± 1.4	119.3 ± 6.9	1.5 ± 0.1	1.0 ± 0.0	1.4 ± 0.1	1.0 ± 0.1
	Severe	477 ± 17.5	42.6 ± 5.8	20 ± 2.4	119.6 ± 5.9	1.5 ± 0.1	1.0 ± 0.0	1.5 ± 0.1	1.0 ± 0.1

## Data Availability

The data presented in this study are available on request from the corresponding author.

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
