# Peer review of "Evaluation of the Cardiometabolic Disorders after Spinal Cord Injury in Mice"

_biology, 2022, doi:10.3390/biology11040495_

Round 1

Reviewer 1 Report

The authors of this study set out to determine the chronic effect of T8 SCI on cardiometabolic function in mice. Using 2 different injury paradigms the investigative team conducted an impressive number of examinations covering multiple body systems as well as motor function. Whilst the manuscript has excellent breadth and offers a rare glimpse into systems physiology in the context of SCI it could benefit from a number of clarifications, a stronger rationale for the injury model, and some additional assays that would advance interpretation. I have outlined these suggestions below:

Rationale: The general rationale for the study of cardiometabolic disease progression is relatively well structured. However, a critical overlooked aspect is the choice of animal model used for this study – the author’s must provide a more compelling rationale for using an injury model that damages right in the middle of the levels that control SNS outflow from the cord (i.e., T8). This level of injury complicates the interpretation of findings as some organs will retain full sympathetic control (i.e., the heart) and some will have partial sympathetic control (i.e., the splanchnic viscera). This matter is further complicated with prior reports of a sympathetic hyper-innervation above the level of injury when injuries are induced at the mid-thoracic level (see the Lujan & DiCarlo manuscripts; PMID: 22723636 & 24610530). To this end the authors need to also rationalize their choice of injury severity – other studies have looked at this with respect to cardiovascular and/or autonomic outcomes so this is not the first study per se (i.e., PMID: 29235182 & 27456150).  

Methods:

1) Were any blood samples also taken from the rats that would enable the investigators to profile either cholesterol/lipids and or plasma NE?

2) Using the short-axis view for echo measurements can result in significant error when inferring volumetric measurements – the authors must use the parasternal long-axis view for these measures. Why were only 3 cardiac-cycles used? This is often the case in human studies but the mouse heart beats significant faster. How did you account for changes in volumes due to the respiratory cycle? Did you gate your analyses to only the expiratory portion? If not, why not? If so, please add this data.

3) Did you also look at insulin levels during the glucose tolerance test?

4) Please provide more information on standardization of the heart and liver sections – i.e., how did you ensure sections were from the same regions across your mice? Which part of the heart? What is the rationale for looking at cardiac fibrosis when control over sympathetic innervation to the heart is intact in this injury model?

5) It is unclear if your sham animals were aged in facility or ordered to age-match the termination point, please clarify.

Results:

1) What did the 4 severely injured mice die from?

2) Figure 2: unclear what the p value comparison group is and which test was used to determine significance, please add this detail to figure legend.

3) Figure 4a – I am not sure I can see a difference between the 2 groups here? What should I be seeing?

4) Can you please explain what indices you are referring to for evidence of cardiac dysfunction? A lower value for LVIDs in the absence of changes in LVIDd would indicate improved emptying, which ties in with your improved ejection fraction. Please also be specific about how your thickness measures in panel 6C were generated? Also, your fibrosis looks like only perivascular fibrosis and not interstitial fibrosis, is this the case?

5) Section 3.6: Can you also stain for the presence of descending sympathetic fibres? i.e., either TH+ and/or 5HT+. This would help understand sparing in autonomic pathways in your injury models and help readers better link the end-organ pathology to injury severity.

Discussion:

A large part of the discussion is spent on cardiac dysfunction, but I do not believe you have any functional evidence of dysfunction. As the authors know, echo at best is limited by  load-dependence of all metrics and no measures of pre-load and/or afterload have been included, making it impossible to discern the presence of cardiac dysfunction.

A key index missing throughout this manuscript is blood pressure – without knowing whether these animals are hypo-/normo-/hyper-tensive it is very difficult to interpret other findings. Are the author’s able to provide any data as to the blood pressure of these mice?               

Conclusion: the authors are advised to stay away from cause and effect here – the inflammation link, whilst certainly plausible, has not been experimentally implicated/demonstrated but rather is associative.

Overall the manuscript could benefit from some English editing to improve sentence structure and clarity.

Author Response

Please refer to the document attached for our in-depth response to the reviewer's comments. 

Reviewer 2 Report

The original article by Ghnenis et al. "Evaluation of the cardiometabolic disorders after spinal cord injury in mice" covers a potentially interesting and emerging topic related to the SCI and its induced pathologies . In this sense, this remains to be potentially interesting for the Biology. I regard the main point of this paper as highly attractive as well as the results are clearly presented. The text does not contain any major errors, therefore I have some minor comments and recommendations:

1.The figure summarizing and clarifying the basic pathogenesis of SCI should be added.

2.Table summarizing number, procedures and experimental groups should be added (for better clarification of results)

3.Following references should be added and properly cited within the main text:

- Tykocki T, Poniatowski ŁA, Czyz M, Wynne-Jones G. Oblique corpectomy in the cervical spine. Spinal Cord. 2018 May;56(5):426-435. doi: 10.1038/s41393-017-0008-4. - Hou S, Rabchevsky AG. Autonomic consequences of spinal cord injury. Compr Physiol. 2014 Oct;4(4):1419-53. doi: 10.1002/cphy.c130045. - Olczak M, Poniatowski ŁA, Niderla-Bielińska J, Kwiatkowska M, Chutorański D, Tarka S, Wierzba-Bobrowicz T. Concentration of microtubule associated protein tau (MAPT) in urine and saliva as a potential biomarker of traumatic brain injury in relationship with blood-brain barrier disruption in postmortem examination. Forensic Sci Int. 2019 Aug;301:28-36. doi: 10.1016/j.forsciint.2019.05.010. - Nash MS, Gater DR Jr. Cardiometabolic Disease and Dysfunction Following Spinal Cord Injury: Origins and Guideline-Based Countermeasures. Phys Med Rehabil Clin N Am. 2020 Aug;31(3):415-436. doi: 10.1016/j.pmr.2020.04.005.

4.In some places the use of English could be improved on.   Completing this gaps will have an impact on the understanding the aim of the study and, from my point of view, is absolutely necessary. The manuscript is timely, the topic is of interest and the study is scientifically sound. I would be in favour of publication after minor correction.

Author Response

(The authors gave the same response as above.)

Round 2

Reviewer 1 Report

The authors have responded adequately to most of my concerns, and provided additional new data which have clearly strengthened the paper. I still have a few comments that I believe have not been fully addressed in the revised version.

  1. The authors still, in my opinion, have not rationalized the choice of the T8 level injury. I had hoped based on my initial suggestions, that the author’s would now include a number of studies in the intro that have in-fact demonstrated that risk for CVD is HIGHER in those with low injuries vs high-level injuries. Though not entirely clear one could surmise that this is probably because of the associated hypoactivity in the sympathetic nervous system in the HIGHER injuries, which prevents many of the maladaptive changes at the organ level and in blood vessels that the SNS causes. The intro, in my opinion, still falls short of a valid clinical rationale for exploring this level of injury and must be strengthened.
  2. The response about the echocardiography analyses is fine, but based on the provided images in the document there is no evidence that you orientated the M mode image from a LONG-AXIS view of the LV. The provided image shows a SHORT-AXIS B-MODE on the top image and the corresponding M-Mode image underneath. The methods must be corrected to reflect this.
  3. RE: Heart wall measures from histology. I had missed that histological images (and not ultrasound) images were used to quantify LV wall thickness in your original submission. How did you ensure the heart was stopped in the same region of the cardiac cycle before perfusion? As you know, the wall thickness changes dramatically between systole and diastole (as can be seen on your M-mode trace). This is critically important given you show change in this metric (in what I think is the wrong direction with injury). Did you assess this on ultrasound too, and if so does the pattern of findings hold consistent?
  4. Thank you for providing the TH data – I think this is a useful addition. Can you confirm what you normalized the density too (if it was %tissue area, did you confirm if that was different between groups)? Also, which part of the LV was this staining performed in? Given the helical nature of the LV fibres, TH+ density changes as the innervation pattern changes throughout the LV.
  5. I am still not sure what changes in systolic function you refer to in lines 591-592? As far as I can tell you have only a change in dimensions, which infers a change in VOLUME. This is likely caused by a SCI-reduction in circulating blood/plasma volume and reduced metabolic demand. As per the first review, you have a reduction in ESV and increase in FS. In the absence of any further data, please be specific in that you have reduction in inferred volumes not systolic function.
  6. Your discussion around LV thickening lacks a physiological mechanism. What could be causing this thickening? Also, I am not sure what you mean by cardiac hypertrophy given you have not histologically examined the cardiomyocytes?
  7. Lines 635-638. This is confusing. What do you mean by blood pressure in the lower part of the body? The arterial system is a closed system…..
  8. The new lipid data is a nice addition. However, to my knowledge blood lipid levels vary considerably in clinical studies, with many showing SCI causes a reduction in lipids. So I think a more balanced discussion is needed here.

Author Response

Please see the attached document for point by point answers to the reviewer's comments
